# A Computer-Driven Approach to Discover Natural Product Leads for Methicillin-Resistant *Staphylococcus aureus* Infection Therapy [note 1]

**DOI:** 10.3390/md17010016

**Published:** 2018-12-28

**Authors:** Tiago Dias, Susana P. Gaudêncio, Florbela Pereira

**Affiliations:** 1UCIBIO-REQUIMTE, Department of Chemistry and Department of Life Sciences, Faculty of Science and Technology, Universidade NOVA de Lisboa, 2829-516 Caparica, Portugal; tiagocalretas@gmail.com (T.D.); s.gaudencio@fct.unl.pt (S.P.G.); 2LAQV-REQUIMTE, Department of Chemistry, Faculty of Science and Technology, Universidade NOVA de Lisboa, 2829-516 Caparica, Portugal

**Keywords:** antibacterial activity, methicillin-resistant *Staphylococcus aureus* (MRSA), quantitative structure–activity relationship (QSAR), machine learning (ML) techniques, molecular descriptors, NMR descriptors, drug discovery, marine natural products (MNPs), marine-derived actinobacteria

## Abstract

The risk of methicillin-resistant *Staphylococcus aureus* (MRSA) infection is increasing in both the developed and developing countries. New approaches to overcome this problem are in need. A ligand-based strategy to discover new inhibiting agents against MRSA infection was built through exploration of machine learning techniques. This strategy is based in two quantitative structure–activity relationship (QSAR) studies, one using molecular descriptors (approach A) and the other using descriptors (approach B). In the approach A, regression models were developed using a total of 6645 molecules that were extracted from the ChEMBL, PubChem and ZINC databases, and recent literature. The performance of the regression models was successfully evaluated by internal and external validation, the best model achieved R^2^ of 0.68 and RMSE of 0.59 for the test set. In general natural product (NP) drug discovery is a time-consuming process and several strategies for dereplication have been developed to overcome this inherent limitation. In the approach B, we developed a new NP drug discovery methodology that consists in frontloading samples with 1D NMR descriptors to predict compounds with antibacterial activity prior to bioactivity screening for NPs discovery. The NMR QSAR classification models were built using 1D NMR data (^1^H and ^13^C) as descriptors, from crude extracts, fractions and pure compounds obtained from actinobacteria isolated from marine sediments collected off the Madeira Archipelago. The overall predictability accuracies of the best model exceeded 77% for both training and test sets.

## 1. Introduction

The use as well as misuse of antibiotics in animal feeding and human medicine has resulted in global antimicrobial resistance epidemics [1,2,3]. The Center for Disease Control and Prevention classifies methicillin-resistant *Staphylococcus aureus* (MRSA) as a severe threat in health care, leading to more than 80,000 invasive infections and resulting in over 11,000 deaths per year [4]. MRSA infection is one of the leading causes of hospital-acquired infections and is commonly associated with significant morbidity, mortality, length of stay, and cost burden in hospital, e.g., the mortality rates vary from 5–60% and depending on the patient population and site of infection [3,5]. Recently was reported an impressive percentage of patients (ca. 60%) acquire MRSA nosocomial infections within 48 h despite having no healthcare issues. New approaches to overcome nosocomial infections and antibiotic resistance are in demand, thus accelerating the discovery of new antibacterial drugs is highly desirable.

Drug research and development (R&D) is comprehensive, complex, expensive, time-consuming, and risky process with clinical success rate of approximately 12% reported in 2014 [6]. In order to shorten the research cycle and reduce the costs of drug R&D, several new methodologies have been developed and applied, e.g., computer-aided drug design (CADD) methods. In the last three decades, CADD methods have emerged as powerful tools in the development of therapeutically important small molecules with higher hit rates than those obtained for more conventional approaches such as high throughput screening (HTS) approaches. More than fourteen FDA-approved drugs were mainly CADD-driven drugs [7], e.g., Norfloxacin (Noroxin^®^) an antibiotic belonging to the class of fluoroquinolone approved in 1986, developed by using a ligand-based CADD approach through quantitative structure–activity relationship (QSAR) methodology [8].

Although target-based approaches using inhibition targets such as ribosome, dihydrofolate reductase (DHFR), β-lactamase, caseinolytic proteases (Clp proteases), fatty acid synthases (FAS I and FAS II), phospholipids in bacterial cell membrane, peptidoglycan biosynthesis enzymes (MurB, C, D, E, and F, MraY, and MurG), transglycosylase of penicillin binding protein, a menaquinone biosynthesis enzyme (MenA), type II NADH oxidoreductase (NDH-2), bacterial DNA polymerase IIIC (Pol IIIC), phosphoinositide-dependent protein kinase-1 (PDK1), phosphatidyl inositol-3-kinase (PI3K), a histidine protein kinase (KinA) and a response regulator (Spo0F), have become major tools for discovering anti-MRSA agents, the pipeline for novel-mode of action anti-MRSA drugs is still empty and will remain that way for a considerable amount of time before a new target-driven drug gets into the market for clinical use [9,10,11]. Therefore alternative approaches should be developed in order to avoid the forthcoming urgent situation of not having appropriate antibiotics for medical needs [3]. Compared to target-based approaches, the traditional whole cell-based screening approach is favored for finding lead compounds with activity against MRSA [9]. A whole-cell assay validates also if the target active agent interaction has anti-MRSA functionality. 

In the last years, some studies have been reported on CADD for antibacterial activity, which used small data sets that are generally focused on a single family of compounds e.g., indolylidinepyrazolone derivatives [12], N_2_-acyl isonicotinic acid hydrazide derivatives [13], (*Z*)-2-(nitroimidazolylmethylene)-3(2H)-benzofuranone derivatives [14], peptides [15,16], anthraquinone derivative [17], indole carboxamide 4-piperazine derivatives [18]. An in silico strategy based on QSAR molecular topology using multi-linear regression (MLR) analysis was reported by Zanni et al. to identify synthetic molecules from 141 heterogeneous active or inactive structural compounds (e.g., benzoimidazole, dihydropirrolone, dihydro-purindione, dinitrobenzene, phenyl sulfonyl morpholine, and indole derivatives) as antimicrobial agents not susceptible to one or several mechanisms of resistance such as biofilm formation, ionophore activity, epimerase activity or SOS system [19]. A QSAR for antimicrobial activity against MRSA based also in molecular topology descriptors was developed by Lopez and co-workers [20] with 56 active and inactive 4-quinolone analogs using MLR and linear discriminant analysis (LDA). This model was successfully applied in the virtual screening of a combinatorial library of more than thousand 6-fluoroquinolone derivatives to the search for new compounds exhibiting significant activity against MRSA. A QSAR approach, for classification, was used by us for the prediction of active/inactive compounds relatively to overall biological activity, antitumor and antibiotic activities using a data set of 1804 compounds from PubChem database [21]. Using the best classification models for antibiotic activity a data set of marine and microbial natural products from the AntiMarin database were screened and 57 new lead compounds for antibiotic drug design were proposed. Using the first rule of the antibiotic tree model, TopoPSA (topological polar surface area) ≥ 120.7 Å^2^ for antibiotic compounds and TopoPSA < 120.7 Å^2^ for non-antibiotic compounds, it was possible to correctly classify 1397 out of 1804 compounds for the training set (ca. 77%) [21]. In spite of the predictive power of the TopoPSA descriptor for modeling the antibiotic activity has never been reported, the high correlations of this descriptor with passive drug transport through membranes [22] and drug transport by multidrug resistance associated protein 1 (MRP1/ABCC1) [23] were already reported. Recently, Ebejer et al. [24] reported the comparison between the physicochemical properties of antibacterial compounds and other drugs and concluded that the percentage of compounds with TopoPSA < 120 Å^2^ is 84.4% and 52.6% for marketed other drugs and antibacterial drugs, respectively, which is in agreement with our previously stablished rule [21]. Wang et al. developed in silico classification models derived from 5451 cell-based anti-MRSA assay data using four machine learning (ML) methods, including naïve Bayesian, support vector machine (SVM), recursive partitioning, and *k*-nearest neighbors [25]. The best model was employed for the virtual screening of anti-MRSA compounds, which were validated by cell-based assays with three types of highly resistant MRSA strains and a total of 12 new anti-MRSA agents were experimentally confirmed by the authors [25]. 

Although the vast majority of currently marketed antibiotics have been isolated from terrestrial microbes, accounting for more than 75% of all antibiotics discovered [26,27], the antibacterial compounds isolated from marine sources have not yet been developed into clinical trials [28,29,30]. However, several novel bioactive compounds with antibacterial activity against MRSA isolated from marine bacteria and fungi were reported [10,30]—(1) isolated from marine-derived bacteria: marinopyrrole A [31], an alkaloid endowed (*Streptomyces* sp.), and abyssomicin C [32], a spirotetronate poliketide (*Verrucosispora* sp.); (2) isolated from marine-derived fungus: spiromastixones A–O [33], depsidone analogs (*Spiromastix* sp.). The marine environment has been proposed as an untapped source of new bioactive molecules, and marine bacteria and fungi seem to be the most important sources for antibacterial compound discovery [29,30,34,35,36]. Computational methodologies such as those using in dereplication and CADD are crucial in the systematic exploration of the biological activity of marine natural products (MNPs) to improve the rate of drug discovery from marine sources [7].

In the current study, two QSAR approaches (A and B) were explored for the prediction of antibacterial activity against human pathogens MRSA, using the value of the minimum inhibitory concentration (MIC). In the approach A was used molecular descriptors for building regression models that predicted the MIC value of antibacterial activity. These models were developed using in total 6645 molecules that were extracted from the ChEMBL, PubChem, and ZINC databases and recent literature indexed in Web of Science. In order to reduce the time consumption and biological activity screening-associated costs that are associated to NP discovery, a new NP drug discovery strategy similarly to that we had recently developed for anticancer activity [37] was developed. In the approach B was used NMR descriptors for building NMR QSAR models to predict compounds with antibacterial activity prior to bioactivity screening for NP discovery. The NMR QSAR classification models were built using 1D NMR data (^1^H and ^13^C) as descriptors, from 50 crude extracts, 55 fractions, and 50 pure compounds obtain from actinobacteria isolated from marine sediments collected off the Madeira Archipelago [38]. The performance of the models was successfully evaluated by internal and external test set validations. Further external validations through data from the StreptomeDB 2.0 [39] database and using MNPs isolated in our MNP research group were also accomplished.

## 2. Results and Discussion

### 2.1. Chemical Space of the Anti-MRSA Models

#### 2.1.1. Approach A

The whole data set of 6645 small molecules was randomly divided into a training set of 5112 molecules (comprising 1688 active molecules with MIC < 5 µM and 3424 inactive molecules with MIC ≥ 5 µM) and a test set of 1533 molecules (comprising 543 active molecules and 990 inactive molecules), which were used for the development and external validation of the QSAR regression models, respectively. The whole data set was clustered into five structural classes or scaffold types (A–E) using the ward tool in JChem. The five structural clusters are represented in Table 1 along with as their average and maximum anti-MRSA pMIC values (i.e., $-\log_{10}(MIC$) in terms of molar concentration).

Although the molecules of these five structural clusters were distributed over a wide range of pMIC values, between 2.64 and 11.82, all clusters showed an average pMIC value greater than or equal to 4.59 (corresponding to a MIC value less than or equal to 25.7 mM), and a maximum pMIC value greater than 7.80 (corresponding to a MIC value less than 0.0158 mM) for the training set. The well-known, Lipinski rule, informs mainly if a molecule is more likely to be an orally administrated active drug and if it is easily absorbed by the body. Furthermore, one of the most important molecular descriptors is the octanol-water partition coefficient (Log *P*), which is highly correlated with lipophilicity, thus, more lipophilic molecules are often discontinued from drug development and are frequently related to toxicity issues [40]. Besides molecular weight (MW) and Log P, as described above in the introduction the electronic descriptor TopoPSA appears to have a remarkable performance in discriminating between antibiotic and non-antibiotic compounds [21]. In order to exploit the training set chemical and biological diversity, the active and inactive molecules of the training set were analyzed, in accordance with the five structural clusters, using MW, XLogP (an estimation of the octanol-water partition coefficient, Log P) and TopoPSA. The analysis of these data indicates that the active and inactive molecules against MRSA in the training set are distributed over a wide range of MW (i.e., 89–1490 Da), XLogP (i.e., −9.34–21.90) and TopoPSA (i.e., 0–644.98 Å^2^) values. Interestingly, approximately 63% of the compounds present in the training set have a MW bellow 500 Da. This MW interval contains approximately 54% and 67% of all active and inactive molecules against MRSA in the training set, respectively. In the same way, using this rule (MW < 500 Da) it is possible to discriminate inactive molecules in relation to active molecules in four structural clusters, namely in clusters: A, B, D, and E, which comprises 67%, 69%, 70%, and 56% of inactive molecules as compared to 48%, 57%, 52%, and 44% of active molecules, respectively. The same behavior was described by Ebejer et al. [24] in which only 75% of marketed antibacterial drugs have a MW bellow 500 Da, whereas for marketed other (non-antibacterial) drugs was 86.7%. In addition, more than 68% and 69% of the active and inactive molecules against MRSA in the training set have an XLogP that is lower than 5, respectively. In spite of this using the XLogP < 5 it is possible to prioritize active molecules in relation to inactive molecules in two clusters, namely in clusters: D and E, which comprises 74% and 81% of active molecules as compared to 67% and 77% of inactive molecules, respectively. Furthermore, using the rule TopoPSA ≥ 120.7 Å^2^ for antimicrobial compounds, it was possible to discriminate approximately 50% and 41% of all active and inactive molecules against MRSA in the training set, respectively. Therefore, using this rule it was possible to prioritize active molecules in relation to inactive antibacterial molecules in all the five clusters, A-E, which comprises 58%, 38%, 40%, 55%, and 73% of active antibacterial molecules as compared to 43%, 36%, 37%, 36%, and 69% of inactive molecules, respectively.

#### 2.1.2. Approach B

The whole data set, covering 155 samples (50 crude extracts, 55 fractions, 50 pure compounds), was divided into a training set of 116 samples (37 crude extracts, 41 fractions, and 38 pure compounds) and a test set of 39 samples (13 crude extracts, 14 fractions, and 12 pure compounds), which were used for the development and external validation of the QSAR classification models, respectively. Two classes of antibacterial activity against MRSA were set, moderate-active-to-active with MIC < 100 µg/mL (in total 49 samples) and inactive with MIC ≥ 100 µg/mL (in total 106 samples). The whole data set comprising five actinobacteria genera (i.e., *Actinomadura*, *Brevibacterium*, *Micromonospora*, *Salinispora*, and *Streptomyces*) in accordance with our previously reported work [38]. The five actinobacteria genera are represented in Table 2 along with their activity classes and average anti-MRSA MIC values. It is interesting to highlight that the most abundant genera in our data set are *Streptomyces* (in total 80 samples, 62 and 18 samples in the training and test sets, respectively), *Salinispora* (in total 39 samples, 29 and 10 samples in the training and test sets, respectively), and *Micromonospora* (in total 32, 23, and 9 samples in the training and test sets, respectively). The genus with the most bioactive potential against MRSA is *Streptomyces*, comprising 28 (corresponding to 45%) and 9 (corresponding to 50%) active samples out of the 62 and 18 total samples in the training and test sets, respectively.

This result is not surprising since the genus *Streptomyces* over the last decades has stirred huge interest as a source of bioactive compounds, more than 60% of all known antibiotics have been isolated from streptomycetes [39], and moreover a similarly result was obtained by us for the anticancer screening against HCT116 in a previously study from actinobacteria isolated from marine sediments collected off the Madeira Archipelago [37].

### 2.2. Exploration of Empirical Molecular Descriptors and Fingerprints for QSAR Approach A

Two wide sets of descriptors were explored, one with 6 different types of fingerprints (FPs) with different sizes (166 MACCS, MACCS keys; 307 Substructure, presence and count Sub and SubC respectively; 881 PubChem fingerprints; 1024 CDK, circular fingerprints; and 1024 CDK Ext, extended circular fingerprints with additional bits describing ring features) and other with a total of 218 1D and 2D descriptors (including 6 electronic, 195 topological, and 17 constitutional descriptors). The FPs and the molecular descriptors were calculated by PaDEL-Descriptor [41] and CDK Descriptor Calculator (http://www.rguha.net/code/java/cdkdesc.html), respectively. Random Forests (RF) [42] ML technique was used for building pMIC against MRSA prediction regression models, and the performance of the models was successfully evaluated by internal validation (out-of-bag, OOB, estimation on the training set), Table 3.

The four fingerprints sets, MACCS, SubC, PubChem and CDK Ext achieved the best results, taking into account the value of the RMSE (Table 3, the best models are highlighted in bold). For us it was very surprising the poor performance of the 1D and 2D descriptors in the modeling of pMIC against MRSA since several molecular descriptors, as detailed in Section 2.1.1, correlate relatively well with the values of pMIC. In this way we analyzed the Pearson correlation (R) of each of these descriptors with the pMIC values, and it was possible to verify that 36, 73, and 126 descriptors out of the 179 descriptors (39 descriptors have constant values) have an absolute R_pMIC_ higher than the R_pMIC_ obtained by the MW (R_pMIC_ = 0.2388), TopoPSA (R_pMIC_ = 0.1760), and XLogP (R_pMIC_ = 0.0439), respectively. From the seven sets of descriptors and fingerprints, only MACCS, SubC, PubChem, and CDK Ext fingerprints were used in further investigations. The 3D descriptors have a well-established relationship with biological activity and are expected to increase both the accuracy and robustness of the predictive models. Several well-established 3D molecular descriptors were exploited, such as 3D CDK descriptors (e.g., BCUT, WHIM, CPSA and geometrical) and Radial Distribution Function (RDF) pair [43] derived 3D descriptors in which the property is the partial atomic charge, Table 4.

Contrary to what would be expected, the predictive power of the anti-MRSA QSAR models does not improve and even worse by including 3D descriptors to the best four sets (MACCS, SubC, PubChem and CDK Ext). Although the model using the CDK Ext FP with 3D CDK descriptors achieved the best results for the training set with R^2^ of 0.54 and RMSE of 0.67 (Table 4), the model using only the CDK Ext FP still has better performance (R^2^ = 0.57 and RMSE = 0.64, Table 3).

Procedure for feature selection based on RF variable importance were applied to the descriptors of models SubC, PubChem, and CDK Ext—Table 5.

The selection of the 150 most important descriptors from the CDK Ext FP set used to build the model with the RF enabled the training of much smaller RF models with even slightly better prediction accuracies (R^2^ = 0.574 and RMSE = 0.635) than the models trained with the whole set of descriptors (1024 descriptors, R^2^ = 0.572 and RMSE = 0.636) for the training set. 

#### 2.2.1. Exploration of Other State-of-the-art Machine Learning (ML) Techniques

A comparison of three ML techniques, RF, support vector machines (SVM), and Gaussian processes (GPs), for building QSAR pMIC models with the 150 most important descriptors using in the model with CDK Ext FPs by the five structural clusters for training and test sets is shown in Table 6 and Figure 1, respectively.

Variation of the ML algorithm could not achieve any consistent improvement of the results obtained with RF for all molecules in the training and test sets taking into account the R^2^ and RMSE values, Table 6 and Figure 1. For all ML models, the best predictions obtained for the structural clusters are obtained from cluster A—indole derivative centroid—and are also better than those obtained for all training and test set taking into account the RMSE value. Interestingly, the worse predictions obtained taking into account the RMSE value for all the clusters in the training and test set were to the cluster C—2-oxazolidone derivative centroid.

Averaged predictions (consensus) obtained by the RF, SVM and GPs models with the 150 most important CDK Ext FPs (CM1), or by the four best set of descriptors, MACCS, SubC, PubChem and CDK Ext (CM2) further improved the results, Table 7.

The best results for the prediction of pMIC against MRSA are achieved with CM2 model with 35% of the structures predicted with a deviation less than 0.5 and 91% of the structures predicted with a deviation less than 1 for the test set. Figure 2 represents the plot of predicted versus experimental pIMIC against MRSA for the test set using the best consensus model, CM2.

#### 2.2.2. Applicability Domain of the pMIC against MRSA Model

The similarity between a molecule of an external data set and all the 5112 molecules in the training set for the regression MRSA model was used to define its applicability domain. The molecules of the training set were mapped on a Kohonen self-organizing map (SOM), using in-house developed software based on JATOON Java applets [44], on the basis of the 512 FPs calculating by GenerFP program in JChem according to the five structural clusters, Table 1. No information about anti-MRSA activity was used. In Figure 3, a trend for clustering according to structural clusters features of the compounds was shown.

After training, the metric of similarity used was the SOM response patterns after normalization, where d(x,n_i_) is the Euclidian distance between the molecular descriptor vector x and n_i_ (represents the centroid vector of the i^th^ SOM neuron). The threshold definition based on the average of SOM distance (ASD) between each molecule of the test set and all the molecules of the training set was accomplished in order to reduce the MAE of the molecules belonging to the applicability domain of the anti-MRSA model and was set in accordance with the mapping of the five structural clusters (A–E) on SOM and MAE obtained in the best RF model (CM2). The applicability domain of the model is defined as containing all molecules of the training set that were mapped as belonging to one of the five clusters on SOM with an ASD lower than 0.4. Therefore, using this threshold for the test set, 1230 molecules belonging to the applicability domain of the anti-MRSA model were obtained, with R^2^ = 0.693, RMSE = 0.586 and MAE = 0.436. However for the molecules of the test set outside the defined applicability domain (i.e., 303 molecules) worse predictions were obtained, with R^2^ = 0.639, RMSE = 0.620 and MAE = 0.479.

#### 2.2.3. Application of the in silico Anti-MRSA Model in Virtual Screening

In the present study, a virtual screening campaign was carried out to search for new lead-like inhibitors against MRSA. The best model, the consensus model—CM2 that is averaged predictions obtained by the four best set of descriptors, MACCS, SubC, PubChem, and CDK Ext, was selected for the virtual screening procedure. The virtual library comprises 3990 molecules from the StreptomeDB 2.0 database [39], an extended compilation of NPs produced by *Streptomyces*. In addition, StreptomeDB 2.0 includes comprehensive background information e.g., host organisms, predicted physicochemical properties, synthesis routes, and biological activities. Although the virtual library includes 1994 molecules with at least one biological activity record, the complete activity spectrum for this data set is not clarified yet.

The best model, CM2, selected 212 virtual hits from the StreptomeDB 2.0 library with a pMIC against MRSA higher than 5.3 (corresponding to a MIC value lower than 5 mM, the cutoff defined by us for antibacterial active molecules against MRSA). From the 212 virtual hits, there are 138 NPs with at least one biological activity record and 112 NPs with antibacterial activity, in which only 4 NPs are reported as being active against MRSA. The 3990 NPs from the StreptomeDB 2.0 library were mapped on SOM in accordance with five structural clusters of the active and inactive molecules against MRSA, Figure 4.

Furthermore, using the threshold defined for the applicability domain of the anti-MRSA model (i.e., ASD < 0.4), it was possible to prioritize the most probable lead-like antimicrobial against MRSA NPs from the StreptomeDB 2.0 library. Therefore, the virtual hits were reduced to 150 compounds, which were ranked by their predicted pMIC, predicted structural clusters and ASD on SOM and are shown in Appendix A. The predicted structural clusters with the best predictive power in accordance with RMSE for the external test set were A, D and E, Figure 1. None of the NPs that were predicted as belonging to the structural cluster D passed the thresholds defined, i.e., pMIC > 5.3 and ASD < 0.4. The list of twelve NPs that were prioritized based upon their ranking is shown in Table 8 and Figure 5.

From the twelve resulting virtual screening hits only the bis-pyrrole derivative, ID 10301, was already presented in our training or test sets (training set Mol3405, Appendix A).

### 2.3. Exploration of NMR Descriptors for QSAR Approach B

Three ranges for ^13^C NMR and four ranges for ^1^H NMR were used to generate the NMR descriptors and were the following, respectively: (1) 1.5 (133 descriptors), 1.0 (200 descriptors), and 0.5 ppm (400 descriptors); and (2) 0.5 (23 descriptors), 0.2 (61 descriptors), 0.1 (120 descriptors) and 0.05 ppm (240 descriptors). Exploratory QSAR experiments using three NMR descriptors sets (with ^1^H NMR descriptors, ^13^C NMR descriptors and combining ^1^H and ^13^C NMR descriptors) were built with RF algorithm to predict two classes of antimicrobial activity against MRSA (i.e., moderate-active-to-active with MIC < 100 µg/mL and inactive with MIC ≥ 100 µg/mL) within a OOB estimation procedure for the training set (Table 9). In Table 9, only the best model of the twelve models, which were trained combining ^1^H and ^13^C NMR descriptors, is represented.

The best model was achieved using 0.5 and 0.2 ppm ranges for ^13^C and ^1^H NMR, respectively, in total 461 NMR descriptors. As we did for approach A, a procedure for feature selection based on RF variable importance were applied to the descriptors of the best ^13^C and ^1^H NMR model, using three packages the 25, 50 and 100 most important descriptors from the 461 ^13^C and ^1^H NMR descriptors. The selection of the 100 most important descriptors used to build the model with the RF enabled the training of much smaller RF models with even better prediction accuracies (Q = 0.80 and MCC = 0.51) than the models trained with the whole set of descriptors (461 descriptors, Q = 0.76 and MCC = 0.38) for the training set. The 100 most important descriptors comprise 23 ^1^H and 77 ^13^C NMR descriptors.

A comparison of three ML techniques, RF, SVM, and Convolutional Neural Network (CNN), for building QSAR classification models with the 100 most important descriptors using in the model with 461 NMR descriptors for training and test sets is shown in Table 10.

Averaged predictions (consensus) obtained by the RF, SVM, and CNN models with the 100 most important ^13^C and ^1^H NMR descriptors (CM_NMR) further improved the results. The two classes of antimicrobial activity against MRSA were predicted for the training and test sets with Q of 0.81 and 0.77, respectively. The MCC were also improved to 0.55 and 0.49 for the training and test sets, respectively.

Moreover, the CM_NMR model was validated with a final prediction set consisting of four pure compounds not used for any task before, which were isolated in our MNP group. The four MNP are two diketopiperazine derivatives, one napyradiomycin derivative and one unknown compound that appears to be a new macrocyclic derivative. The chemical structure of these 4 MNP has not yet been fully elucidated and consequently they are excellent candidates for this QSAR approach B model. In Table 11 is showed the predictions obtained using the CM_NMR model for the four pure compounds and anti-MRSA experimental assay results.

Only a misclassification between moderate-active-to-active and inactive class was obtained for the pure compound, PTM-420 F5,F45, however its MIC value is close to the threshold for the inactive class, MIC ≥ 100 μg/mL.

#### Analysis of NMR Descriptors Identified as Relevant for Modeling Anti-MRSA Activity in the RF Model

The 100 most important descriptors, found by the RF algorithm using the MeanDecreaseAccuracy parameter (Mean Decrease in Accuracy) [45] and using for building the CM_NMR were analyzed and the fifteen most relevant descriptors were represented in Table 12.

Interestingly, there are nine descriptors that codify ^1^H NMR data out of the ten most important NMR descriptors. From those nine ^1^H NMR descriptors, five descriptors codify polar functional groups such as –NH (3, 8, 19 and 22), –OH (3, 8, 19 and 22), –COOH (19 and 22), C=N–OH (22), or –CHO (28). On the other hand, one out nine ^1^H NMR descriptors codifies saturated alkyl groups (58). The only ^13^C NMR descriptor in the ten most important NMR descriptors discriminated aromatic, olefinic or nitrile functional groups (318). Moreover, it is also verified that importance by activity classes (moderate-active-to-active and inactive classes) for each of the ten most important descriptors is more or less similar except to descriptors C318, which gives the inactive class a higher weight.

## 3. Materials and Methods

### 3.1. Data Sets

#### 3.1.1. Approach A

A data set of 7744 organic molecules was extracted from the ChEMBL (https://www.ebi.ac.uk/chembl/) [46], PubChem (http://pubchem.ncbi.nlm.nih.gov) [47], ZINC (https://zinc15.docking.org/) [48] databases, searching by antibacterial activity against MRSA with MIC values and their chemical structures saved in the SMILES or MDL SDF data format. In addition, more 653 chemical structures with anti-MRSA activity records were also added by searching in the literature indexed in Web of Science™ Core Collection between May 2013 and March 2015. After assembling these databases, the duplicates were removed based on InChI codes, although the chirality was taken into account, racemic molecules (or cases where the stereochemistry was not shown) were considered as one of the possible stereoisomers. For the duplicates with different MIC values, the most recent were considered. After this, the final data set comprises 6645 molecules and the MIC values were converted to pMIC. The SMILES strings of the data set, the corresponding experimental and predicted activities are available as Appendix A.

#### 3.1.2. Approach B

A total of 155 samples was used for approach B, comprising 50 crude extracts, 55 fractions, and 50 pure compounds obtained from microbial actinobacteria isolated from ocean sediments collected off the Madeira Archipelago [38], corresponding to 49 moderate-active-to-active (MIC < 100 µg/mL) and 106 inactive (MIC ≥ 100 µg/mL) samples antibacterial against MRSA. Actinobacteria strains were isolated from the marine sediments, taxonomically characterized through 16S rRNA gene sequencing, and the crude extracts were obtained through liquid–liquid extraction with ethyl acetate (EtOAc) in accordance with our previously reported work [38]. The EtOAc crude extracts were fractionated by silica flash chromatography, eluted with step gradients of isooctane/EtOAc followed by EtOAc/MeOH and were obtained nine fractions. Pure compounds were isolated by reversed phase HPLC (Phenomenex Luna, 250 × 100 mm, 5 µm, 100 Å, 1,5 mL/ min, UV 210, 250 and 360 nm) using a gradient solvent system of acetonitrile and water. The code, type, and the actinobacteria genus of the samples comprising the data set, the corresponding experimental and predicted activity classes are available as Appendix A.

### 3.2. Descriptors

#### 3.2.1. Approach A

The molecular structures were standardized by normalizing tautomeric and mesomeric groups and by removing small disconnected fragments using JChem Standardizer tool version 5.7.13.0 (ChemAxon Ltd., Budapest, Hungary). Three-dimensional models of the molecular structures were generated with CORINA version 2.4 (Molecular Networks GmbH, Erlangen, Germany. Empirical Molecular fingerprints were calculated by PaDEL-Descriptor version 2.21 (Yap Chun Wei, Pharmaceutical Data Exploration Laboratory)) [41]. Different types of fingerprints with different sizes were calculated and explored: 166 MACCS (MACCS keys), 307 Substructure (presence and count of SMARTS patterns for Laggner functional group classification—Sub and SubC respectively), 881 PubChem fingerprints (ftp://ftp.ncbi.nlm.nih.gov/pubchem/specifications/pubchem_fingerprints.txt), 1024 CDK (circular fingerprints), and 1024 CDK extended (Ext circular fingerprints with additional bits describing ring features). 1D, 2D, and 3D Molecular descriptors were calculated by CDK Descriptor Calculator version 1.4.6 (Rajarshi Guha), which comprise 218 1D/2D descriptors (including 6 electronic, 195 topological, and 17 constitutional descriptors) and 66 3D descriptors (including 6 BCUT, 13 WHIM, 29 CPSA and 18 geometrical descriptors). Radial Distribution Function (RDF) pair descriptors [43], 3D RDF descriptors, were calculated by sampling the function of Equation (1) at 128 equally distributed values of r between 0 and 12.8 Å:(1)$$RDF(r)=\sum_{i=1}^{N-1}\sum_{j=i+1}^{N}p_{i}p_{j}e^{-B{\left(r-r_{ij}\right)}^{2}}$$
where *N* is the number of atoms in the molecule, *pi* is the charge of atom *i*, B is a fuzziness parameter (it was 100 in this study), and *r_ij_* is the 3D distance between atoms *i* and *j*. Three sets of 128 RDF descriptors were separately calculated, derived from atom pairs with (a) a positive and a negative charge, (b) two positive charges, and (c) two negative charges. The partial atomic charges—natural bond orbital (NBO) partial atomic charges were estimated using a ML tool developed by Aires-de-Sousa and co-workers [49] (http://joao.airesdesousa.com/charges).

#### 3.2.2. Approach B

All samples were evaluated for antibacterial activity against MRSA and the 1D NMR spectra were also acquired. NMR spectra were obtained using a Bruker Advance spectrometer, model ARX 400, (400 MHz for ^1^H and 100 MHz for ^13^C) with tetramethylsilane (TMS) as internal reference and deuterated chloroform as solvent. NMR spectra were handled with the ACD/NMR Processor (version 12.01) and the range of chemical shifts used was 0–200 ppm and 0–12 ppm for the ^13^C and ^1^H, respectively. The NMR descriptors were generated using the following ranges: (1) 1.5 (133 descriptors), 1.0 (200 descriptors), and 0.5 ppm (400 descriptors) for ^13^C NMR; and (2) 0.5 (32 descriptors), 0.2 (61 descriptors), 0.1 (120 descriptors) and 0.05 (240 descriptors) ppm for ^1^H NMR.

### 3.3. Selection of Training and Test Sets

#### 3.3.1. Approach A

The whole data set of 6645 small molecules was randomly divided into a training set of 5112 molecules (comprising 1688 active molecules with MIC < 5 µM and 3424 inactive molecules with MIC ≥ 5 µM) and a test set of 1533 molecules (comprising 543 active molecules and 990 inactive molecules) in order to the biological diversity of the data set was captured by both sets. The built QSAR models were developed and externally validated using the training and test sets, respectively.

#### 3.3.2. Approach B

The whole data set, comprising 155 samples (50 crude extracts, 55 fractions, 50 pure compounds), was split into a training set of 116 samples (37 crude extracts, 41 fractions, and 38 pure compounds) and a test set of 39 samples (13 crude extracts, 14 fractions, and 12 pure compounds), which were used for the development and external test validation of the QSAR models, respectively. The approximate 3:1 partition for training and test sets, respectively, was carried out randomly according to the two classes of antibacterial activity against MRSA (i.e., moderate-active-to-active with MIC < 100 µg/mL, in total 49 samples, and inactive with MIC ≥ 100 µg/mL, in total 106 samples), the type of sample (i.e., crude extracts, fractions, or pure compounds), and the five actinobacteria genera (i.e., *Actinomadura*, *Brevibacterium*, *Micromonospora*, *Salinispora*, and *Streptomyces*) in order to the biological diversity of the data set was captured by both sets.

### 3.4. Machine Learning (ML) Techniques

#### 3.4.1. Random Forests (RF)

A RF [42,50] is an ensemble of unpruned trees, which are created using bootstrap samples of the training set and for each individual tree the best split at each node is defined using a randomly selected subset of descriptors. A different training and validation set was used to create each individual tree. Prediction is made by a majority vote of the classification trees (classification) or by average of the individual regression trees (regression) in the forest. The prediction error for the compounds left out in the bootstrap procedure (internal cross-validation or OOB estimation) was used to assess the internal performance. The RF method quantifies also the importance of a descriptor by the increase in misclassification occurring when the values of the descriptor are randomly permuted, correlated with the mean decrease in accuracy parameter, or by the decrease in a node’s impurity every time the descriptor is used for splitting, correlated with the mean decrease in the Gini coefficient parameter. In the experiments presented here, RF were used for the development of regression or classification models to estimate antibacterial activity against MRSA. The R program [51], version 3.2.3 was used to grow RF using the RandomForest library [52]. The number of trees in the forest was set to 500 and the other parameters, except mtry, were used with default values. The mtry parameter values were selected using factor levels of the default value (i.e., 1/3 of the number of descriptors or square root of the number of descriptors in the data for regression or classification, respectively).

#### 3.4.2. Support Vector Machines (SVM)

SVM [53] map the data into a hyperspace through a nonlinear mapping (a boundary or hyperplane), for classification models the two class of compounds are separated in this space and for regression models a linear regression is performed in this space. Support vectors, which are examples in the training set, were used to position the boundary. Kernel functions can be used to transform nonlinear data into a hyperspace where the classes become linearly separable. In the current study, SVM models were explored with the Weka (version 3.8.2) [54] implementation of the LIBSVM software [55]. The C-SVM-classification or ε-SVM-regression types were chosen, the kernel function selected was the radial basis function and used the default value for the gamma parameter, and the parameter C was optimized in the range of 1–200 through 10-fold cross-validation with the training set.

#### 3.4.3. Gaussian Processes (GPs)

GPs use lazy learning and a measure of the similarity between compounds (the kernel function) to predict the value for an unseen compound from training set. The prediction is not just an estimate for that compound, but also has uncertainty information, it is a one-dimensional Gaussian distribution. It was used in this work as specifically implemented by the Gaussian Processes class in Weka version 3.8.2 [54], with the options set as default, except the kernel function and the noise that were optimized in cross-validation experiments with the training set. GPs were implemented for regression without hyperparameter-tuning using a RBF kernel, Equation (2).
(2)$$K\left(x,y\right)=e^{-\gamma {\left(x-y\right)}^{2}}$$

The noise level was chosen applying normalization/standardization to the target attribute as well as the other attributes based on normalize training set.

#### 3.4.4. Convolutional Neural Network (CNN)

CNN is a feed-forward neural network (NN) with more than one hidden layers or convolutional layers. CNN implementation with dropout regularization and Rectified Linear Units. The Weka [54] NeuralNetwork package (version 3.8.2) was used to implement feed-forward neural networks. The NeuralNetwork options were set as default, except the number of hidden units, number of layers, hiddenLayersDropoutRate and inputLayerDropoutRate parameters that were optimized in cross-validation experiments with the training set.

### 3.5. Antibacterial Screening against MRSA Strain

The antibacterial activity was evaluated for the crude extracts, fractions and pure compounds by performing screening against methicillin-resistant *Staphylococcus aureus* (MRSA) COL [56], using Brain Heart Infusion (BHI) medium (DIFCO Laboratories, Detroit, USA, 1l DI water). The screening was performed in 96 well plates; each crude extract, fractions or pure compound, previously concentrated at 10mg/mL in DMSO, was added to a log-phase grown culture (OD_600_ nm = 0.04−0.06) to a 2.5% (*v*/*v*) final concentration. All samples were two-fold serially diluted five times, resulting in final concentrations of the tested compounds ranging from 250 to 7.81 µg/mL. Further dilutions were tested, values down to 0.03 µg/mL, if the 7.81 µg/mL concentration showed inhibitory activity. After 18 h of incubation at 37 °C, minimal inhibitory concentrations were determined by visual inspection and spectrophotometric analysis.

## 4. Conclusions

Following our previous work that modeling the anticancer activity against HCT116 [37], the current results suggest as well that the chemoinformatics QSAR approach relying on a ligand-based methodology either based on the molecular structures or the NMR spectra, corroborated with an experimental approach, could be used to predict new inhibitory compounds against MRSA. To our knowledge, the QSAR regression model developed here, approach A, is the largest study ever performed with regard both to the number of compounds involved and to the number of structural families involved in the modeling of the antibacterial activity against MRSA [19,20,25]. The NMR QSAR classification model, approach B, was extended to a high number of samples containing additional 45 pure compounds and therefore the overall predictability accuracies (Q) were improved as compared with those obtained in our previously work [37]. The Q of the best model exceeded 77% and 63% for test set in MRSA and HCT116 modeling, respectively. Moreover, the development of QSAR model using predicted NMR spectra of molecules and mixture of molecules (simulating fractions and extracts) would be an important strategy to be developed in future work.

## Figures and Tables

**Figure 1 marinedrugs-17-00016-f001:**
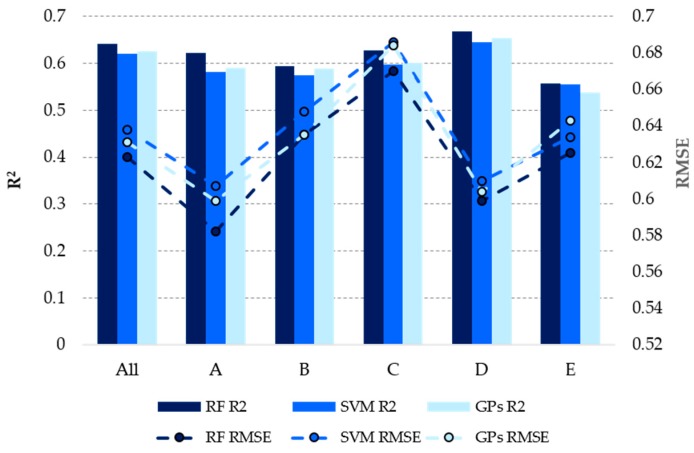
Performance of different ML algorithms by the five structural clusters for the test set.

**Figure 2 marinedrugs-17-00016-f002:**
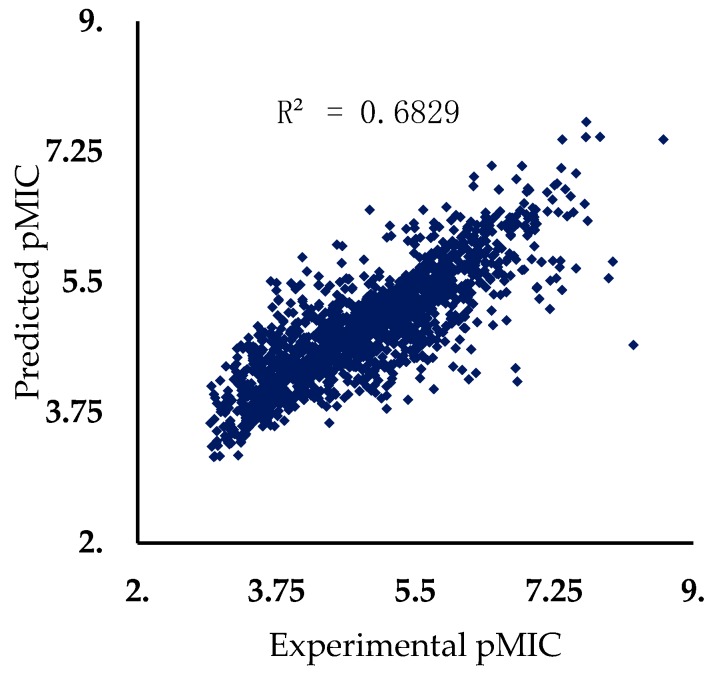
Predicted versus experimental pMIC against MRSA for the 1533 molecular structures of the test set.

**Figure 3 marinedrugs-17-00016-f003:**
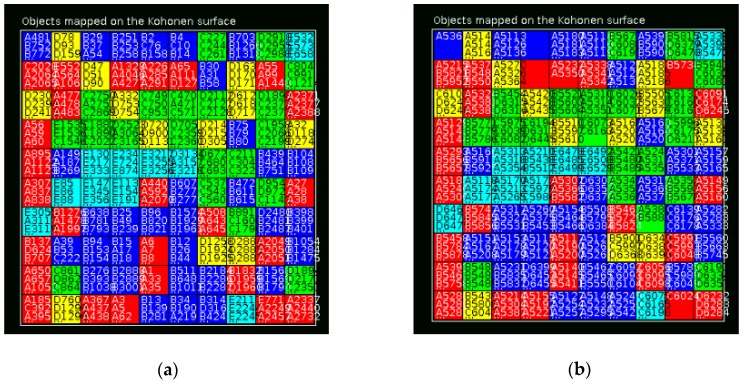
Mapping of the trained and predicted structural clusters of the active and inactive molecules against MRSA on self-organizing map (SOM) for the: (**a**) Training set; (**b**) Test set. Red—cluster A, dark blue—cluster B, green—cluster C, light yellow—cluster D, light blue—cluster E.

**Figure 4 marinedrugs-17-00016-f004:**
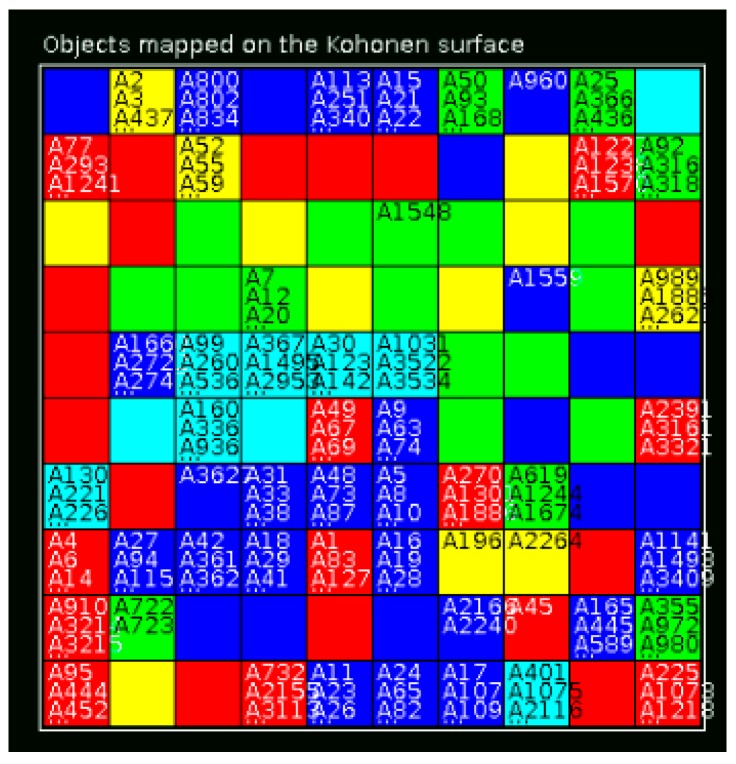
Mapping of the predicted structural clusters of the active and inactive molecules against MRSA on SOM for the StreptomeDB 2.0 library. Red—cluster A, dark blue—cluster B, green—cluster C, light yellow—cluster D, light blue—cluster E.

**Figure 5 marinedrugs-17-00016-f005:**
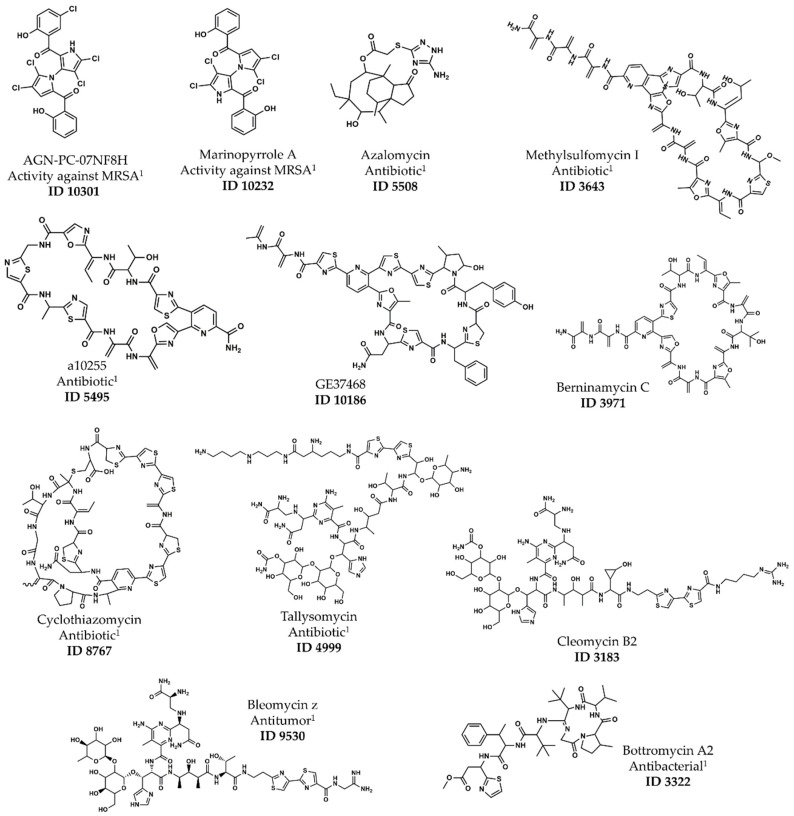
Chemical structures of the twelve resulting virtual screening hits. ^1^ Biological activity reported in StreptomeDB 2.0.

**Table 1 marinedrugs-17-00016-t001:** Structural clusters and pMIC values for anti-MRSA within the five clusters.

Clusters ^1^	Training Set ^2^	Test Set ^2^	Average/MaximumpMIC ^3^
**A**—Indole derivative 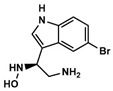	1123	324	4.72/7.80
**B**—1H-2-Benzopyran derivative 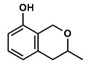	1657	517	4.59/7.81
**C**—2-Oxazolidone derivative 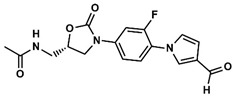	879	253	5.09/11.82
**D**—Triazole derivative 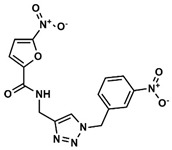	913	270	5.12/9.00
**E**—Cephalosporin derivative 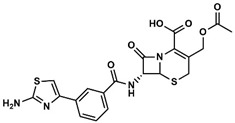	540	169	5.18/8.68

^1^ Cluster number and chemical structure of the cluster centroid. ^2^ Number of molecules. ^3^ Within the cluster for the training set.

**Table 2 marinedrugs-17-00016-t002:** Actinobacteria genera and correspondent anti-MRSA MIC values.

Actinobacteria Genera	Set (Number/Sample Types)	Activity Class/Average MIC ^1^
*Actinomadura*	Tr set ^2^ (2, cr ^4^)	cr: inactive/>250
	Te set ^3^ (1, cr ^4^)	cr: inactive/>250
*Brevibacterium*	-	-
	Te set ^3^ (1, cr ^4^)	cr: inactive/>250
*Micromonospora*	Tr set ^2^ (23, 3 cr ^4^, 13 fr ^5^, 7 pu ^6^)	cr: inactive/>250 fr: 5 active/31; 8 inactive/≥250 pu: inactive/>250
	Te set ^3^ (9, 3 cr ^4^, 3 fr ^5^, 3 pu ^6^)	cr: 2 active/35; 1 inactive/>250 fr: 1 active/8; 2 inactive/>250 pu: inactive/>250
*Salinispora*	Tr set ^2^ (29, 11 cr ^4^, 10 fr ^5^, 8 pu ^6^)	cr: 1 active/63; 10 inactive/>250 fr: 2 active/23; 8 inactive/≥250 pu: inactive/>250
	Te set ^3^ (10, 4 cr ^4^, 4 fr ^5^, 2 pu ^6^)	cr: inactive/>250 fr: 1 active/8; 3 inactive/>250 pu: inactive/>250
*Streptomyces*	Tr set ^2^ (62, 21 cr ^4^, 18 fr ^5^, 23 pu ^6^)	cr: 2 active/5; 19 inactive/243 fr: 12 active/11; 6 inactive/229 pu: 14 active/12; 9 inactive/222
	Te set ^3^ (18, 4 cr ^4^, 7 fr ^5^, 7 pu ^6^)	cr: inactive/219 fr: 5 active /23; 2 inactive/188 pu: 4 active/11; 3 inactive/188

^1^ μg/mL. ^2^ Training set. ^3^ Test set. ^4^ Crude extracts. ^5^ Fractions. ^6^ Pure compounds.

**Table 3 marinedrugs-17-00016-t003:** Exploration of two collections of empirical descriptors for the quantitative structure–activity relationship (QSAR) Random Forests (RF) model of pMIC against MRSA for the training set with an OOB estimation.

Descriptors (#)	R^2^	RMSE ^1^	MAE ^2^	% Error ≥ 0.5/% Error < 0.5 ^3^
MACCS (166) ^4^	**0.555**	**0.649**	**0.477**	**37/63**
Sub (307) ^4^	0.509	0.681	0.505	39/61
SubC (307) ^4^	**0.563**	**0.642**	**0.468**	**36/64**
PubChem (881) ^4^	**0.561**	**0.643**	**0.467**	**36/64**
CDK (1024) ^4^	0.551	0.652	0.471	36/64
CDK Ext (1024) ^4^	**0.572**	**0.636**	**0.456**	**34/66**
1D2D (218) ^5^	0.364	0.775	0.600	49/51

^1^ Root mean squared error. ^2^ Mean absolute error. ^3^ Percent of molecules predicted with absolute error above or below 0.5. ^4^ Fingerprints. ^5^ Molecular descriptors.

**Table 4 marinedrugs-17-00016-t004:** Exploration of 3D descriptors for building anti-MRSA RF models with the best four set of fingerprints (FPs) (MACCS, SubC, PubChem, and CDK Ext) for the training set in an OOB estimation.

Descriptors (#)	R^2^	RMSE	MAE	% Error ≥ 0.5/% Error < 0.5 ^1^
MACCS_3D_CDK (232) ^2^	0.441	0.731	0.569	47/53
MACCS_3D_RDF (550) ^3^	0.426	0.745	0.587	45/55
SubC_3D_CDK (373) ^2^	0.466	0.715	0.556	47/53
SubC_3D_RDF (691) ^3^	0.448	0.730	0.574	48/52
PubChem_3D_CDK (947) ^2^	0.479	0.705	0.546	45/55
PubChem_3D_RDF (1265) ^3^	0.465	0.719	0.564	48/52
CDK_Ext_3D_CDK (1090) ^2^	0.537	0.667	0.507	41/59
CDK_Ext_3D_RDF (1408) ^3^	0.507	0.690	0.533	44/56

^1^ Percent of molecules predicted with absolute error above or below 0.5. ^2^ Comprising in total 66 descriptors, 6 3D-BCUTS, 29 CPSA, 9 Gravitational index, 7 Moment of inertia, 2 Petitjean shape index, and 13 WHIM. ^3^ RDF Pair calculated in sets of charge pairs covering a distance of 12.8 Å, obtaining in total 384 descriptors.

**Table 5 marinedrugs-17-00016-t005:** RF Prediction of pMIC against MRSA with subsets of descriptors from models SubC, PubChem, and CDK Ext.

Model/n.° Descriptors	R^2^	RMSE	MAE	% Error ≥ 0.5/% Error < 0.5 ^1^
Training set ^2^				
SubC/75 ^3^	0.556	0.647	0.471	36/64
SubC/100 ^3^	**0.563**	**0.642**	**0.467**	**36/64**
SubC/150 ^3^	0.562	0.643	0.467	36/64
Pubchem/75 ^3^	0.528	0.667	0.496	39/61
PubChem/100 ^3^	0.544	0.656	0.484	38/62
PubChem/150 ^3^	**0.558**	**0.645**	**0.471**	**37/63**
CDK Ext/75 ^3^	0.545	0.655	0.481	37/63
CDK Ext/100 ^3^	0.566	0.641	0.467	36/64
CDK Ext/150 ^3^	**0.574**	**0.635**	**0.460**	**35/65**
Test set				
SubC/75 ^3^	0.637	0.626	0.464	36/64
SubC/100 ^3^	0.645	0.620	0.459	36/64
SubC/150 ^3^	0.644	0.621	0.460	36/64
Pubchem/75 ^3^	0.603	0.654	0.494	39/61
Pubchem/100 ^3^	0.620	0.639	0.475	37/63
Pubchem/150 ^3^	0.632	0.629	0.471	37/63
CDK Ext/75 ^3^	0.609	0.650	0.483	38/62
CDK Ext/100 ^3^	0.632	0.631	0.467	37/63
CDK Ext/150 ^3^	0.641	0.623	0.458	34/66

^1^ Percent of molecules predicted with absolute error above or below 0.5. ^2^ OOB estimation. ^3^ Using the mean decrease in accuracy measure of importance for the descriptors in the RF algorithm.

**Table 6 marinedrugs-17-00016-t006:** Performance of different machine learning (ML) algorithms by the five structural clusters for the training set. The models comprising all the molecules of training set are highlighted in bold.

ML		R^2^	RMSE	MAE
RF ^1^	**All**	**0.574**	**0.635**	**0.460**
	A	0.583	0.555	0.416
	B	0.549	0.622	0.458
	C	0.445	0.757	0.520
	D	0.594	0.641	0.467
	E	0.575	0.600	0.449
SVM ^2^	**All**	**0.564**	**0.645**	**0.457**
	A	0.584	0.556	0.407
	B	0.518	0.649	0.463
	C	0.466	0.743	0.508
	D	0.569	0.659	0.465
	E	0.570	0.606	0.443
GPs ^2^	**All**	**0.568**	**0.638**	**0.465**
	A	0.590	0.548	0.415
	B	0.528	0.636	0.466
	C	0.450	0.752	0.530
	D	0.583	0.647	0.474
	E	0.577	0.599	0.442

^1^ OOB estimation for the training set. ^2^ Ten-fold cross-validation for the training set.

**Table 7 marinedrugs-17-00016-t007:** Performance of the consensus models (CM1 and CM2) predicting pMIC against MRSA for the training and test sets.

Model	R^2^	RMSE	MAE	% Error ≥ 0.5/% Error < 0.5 ^1^
Training set				
CM1 ^2^	0.587	0.624	0.450	33/67
CM2 ^3^	0.601	0.617	0.453	34/66
Test set				
CM1	0.644	0.617	0.453	33/67
CM2	0.683	0.593	0.444	35/65

^1^ Percent of molecules predicted with absolute error above or below 0.5. ^2^ Averaged predictions obtained by the RF, SVM and GPs models with the 150 most important CDK Ext FPs with OOB estimation and ten-fold cross-validation for the training set using RF and the other ML techniques, respectively. ^3^ Averaged predictions obtained by the four best set of descriptors, MACCS, SubC, PubChem and CDK Ext for the training set in an OOB estimation.

**Table 8 marinedrugs-17-00016-t008:** The Twelve resulting virtual screening hits from the StreptomeDB 2.0 library.

ID ^1^	Name	Type	pMIC ^2^	Cluster ^3^	ASD
10301	AGN-PC-07NF8H	Bis-pyrrole	6.06	A	0.23
10232	Marinopyrrole A	Bis-pyrrole	5.92	A	0.23
5508	Azalomycin	Spiro-tricyclic	5.51	A	0.39
3643	Methylsulfomycin I	Pyridine-containing ^4^	7.08	E	0.33
5495	a10255	Pyridine-containing ^4^	6.51	E	0.31
10186	GE37468	Pyridine-containing ^4^	6.41	E	0.35
3971	Berninamycin C	Pyridine-containing ^4^	6.23	E	0.38
8767	Cyclothiazomycin	Polythiazole-containing ^5^	6.02	E	0.34
4999	Tallysomycin	Glycopeptide	5.42	E	0.39
3183	Cleomycin B2	Glycopeptide	5.36	E	0.37
9530	Bleomycin z	Glycopeptide	5.32	E	0.39
3322	Bottromycin A2	Macrocyclic peptide	5.30	E	0.31

^1^ StreptomeDB ID number. ^2^ Predicted pMIC. ^3^ Estimate structural cluster on SOM. ^4^ Thiopeptide. ^5^ Peptide.

**Table 9 marinedrugs-17-00016-t009:** Exploration of three collections of NMR descriptors for the QSAR RF model of antibacterial activity against MRSA classes for the training and test sets.

	Model	# ^1^	TP ^2^	TN ^3^	FP ^4^	FN ^5^	SE ^6^	SP ^7^	Q ^8^	MCC ^9^
Training set ^10^										
^13^C	0.5	400	12	72	8	24	0.33	0.90	0.72	0.29
	1	200	9	71	9	27	0.25	0.89	0.69	0.18
	1.5	133	12	72	8	24	0.33	0.90	0.72	0.29
^1^H	0.05	240	10	73	7	26	0.28	0.91	0.72	0.25
	0.1	120	12	72	8	24	0.33	0.90	0.72	0.29
	0.2	61	14	71	9	22	0.39	0.89	0.73	0.32
	0.5	23	8	69	11	28	0.22	0.86	0.66	0.11
^13^C ^1^H	0.5 0.2	461	13	75	5	23	0.36	0.94	0.76	0.38
Test set										
^13^C	0.5	400	4	24	2	9	0.31	0.92	0.72	0.30
	1	200	2	25	1	11	0.15	0.96	0.69	0.20
	1.5	133	1	22	4	12	0.08	0.85	0.59	0.11
^1^H	0.05	240	7	22	4	6	0.54	0.85	0.74	0.40
	0.1	120	7	21	5	6	0.54	0.81	0.72	0.36
	0.2	61	8	21	5	5	0.62	0.81	0.74	0.42
	0.5	23	7	22	4	6	0.54	0.85	0.74	0.40
^13^C ^1^H	0.5 0.2	461	7	24	2	6	0.54	0.92	0.79	0.52

^1^ Number of descriptors. ^2^ True positives. ^3^ True negatives. ^4^ False positives. ^5^ False negatives. ^6^ Sensitivity, the ratio of true positives to the sum of true positives and false negatives. ^7^ Specificity, the ratio of true negatives to the sum of true negatives and false positives. ^8^ Overall predictive accuracy, the ratio of the sum of true positives and true negatives to the sum of true positives, true negatives, false positives and false negatives. ^9^ Matthews correlation coefficient. ^10^ OOB estimation.

**Table 10 marinedrugs-17-00016-t010:** Exploration of different ML algorithms in the prediction of two classes of antibacterial activity against MRSA using the 100 most important descriptors for the training and test sets.

ML	SE ^1^	SP ^2^	Q ^3^	MCC ^4^
Training set				
RF ^5^	0.56	0.91	0.80	0.51
SVM ^6^	0.72	0.81	0.78	0.52
CNN ^6^	0.61	0.89	0.80	0.52
Test set				
RF	0.46	0.92	0.77	0.45
SVM	0.69	0.73	0.72	0.41
CNN	0.62	0.81	0.74	0.42

^1^ Sensitivity. ^2^ Specificity. ^3^ Overall predictive accuracy. ^4^ Matthews correlation coefficient. ^5^ OOB estimation. ^6^ Ten-fold cross-validation.

**Table 11 marinedrugs-17-00016-t011:** Prediction of activity classes against MRSA of the four pure compounds with the CM_NMR model.

Code	Actinobacteria Genera	Structural Family	Activity Class ^1^	MIC (μg/mL) ^2^	Activity Class ^2^
PTM-290 F7,F26	*Salinispora*	Diketopiperazine	InAct ^3^	>250	InAct ^3^
PTM-290 F7,F27	*Salinispora*	Diketopiperazine	InAct ^3^	>250	InAct ^3^
PTM-420 F4,F15	*Streptomyces*	Unknown	InAct ^3^	>250	InAct ^3^
PTM-420 F5,F45	*Streptomyces*	Napyradiomycin	InAct ^3^	62.5	MAct ^4^

^1^ Predicted. ^2^ Experimental. ^3^ Inactive. ^4^ Moderate-active-to-active.

**Table 12 marinedrugs-17-00016-t012:** Analysis of NMR descriptors for modeling anti-MRSA activity.

H or C (# ^1^)	NMR Range (ppm)	Ranking ^2^	Importance for Classes		Pattern Identification
			InAct ^3^	MAct ^4^	
H (19)	11.2393–11.5676	1st	9.23	6.59	Hydrogen bond CO and –NH and –OH; heteroaromatic NH; COOH
H (8)	13.8656–14.1939	2nd	6.22	4.86	Hydrogen bond CO and –NH and –OH; heteroaromatic NH
H (22)	10.5828–10.9111	3rd	3.92	4.86	Hydrogen bond CO and –NH and –OH; heteroaromatic NH; COOH; C=N–OH
H (28)	9.5979–9.9262	8th	2.30	2.46	Aldehyde CHO
C (318)	127.4927–127.9927	9th	3.08	−0.02	Aromatic; olefinic; nitrile
H (58)	1.3909–1.7191	10th	2.91	1.77	Saturated alkane
H (48)	7.3000–7.6282	11th	2.17	1.05	Aromatic; conjugated olefinic
C (410)	175.9927–176.4927	14th	1.85	0.08	COX; X: O, N, Cl a,b unsat. COX; X: O, N, Cl
C (321)	33.9927–34.4927	15th	2.61	0.58	–CH_2_COX; –NHCH_3_; CH_3_CH_2_CH_2_–
C (350)	168.9927–169.4927	16th	1.97	0.43	COX; X: O, N, Cl,b unsat. COX; X: O, N, Cl
H (36)	5.6585–5.9868	18th	2.66	0.48	Vinylic
C (141)	52.4927–52.9927	19th	2.24	1.00	–CHCl; –CH_2_Cl
C (329)	123.9927–124.4927	20th	1.86	0.93	Vinylic
C (415)	171.9927–172.4927	21th	2.14	3.40	COX; X: O, N, Cl,b unsat. COX; X: O, N, Cl
C (401)	178.4927–178.9927	23th	2.57	0.18	COX; X: O, N, Cl,b unsat. COX; X: O, N, Cl

^1^ Number of descriptor. ^2^ Descriptor importance. ^3^ Inactive. ^4^ Moderate-active-to-active.

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
