# Peer review of "A Computer-Driven Approach to Discover Natural Product Leads for Methicillin-Resistant Staphylococcus aureus Infection Therapy"

_marinedrugs, 2018, doi:10.3390/md17010016_

Reviewer 1 Report

Dias et. al., developed QSAR approaches utilizing molecular descriptors and NMR descriptors. They validated the methods against datasets with antibacterial activity against MRSA. I recommend the article for publication after addressing the following issues. I suggest authors to look for similar issues across the manuscript.

Consider revising title - One of your data sets used is not  Natural product (e.g. Approach A). Also

change Discovery to discover

Line 24: NP hit drug discovery - remove 'hit'

Lines 148-181: Can this be summarized into a table?

Why didn't you use the same datasets for both approaches and do a comparison?

Lines 71-75: One more example is 'indolylidinepyrazolones' Tetrahedron Letters Volume 55, Issue 36, 3 September 2014, 5014-5018

Line 110: check [10,29]-1)

Author Response

Manuscript ID: biomolecules-321595

Title: "A Computer-Driven Approach to Discovery Marine Natural Product Leads for Methicillin-resistant Staphylococcus aureus Infection Therapy "

Author(s): Tiago Dias, Susana P. Gaudêncio and Florbela Pereira

Reviewer 1

Critical

I – Title

Consider revising title - One of your data sets used is not Natural product (e.g. Approach A). Also change Discovery to discover

 In the title “Discovery” was changed to “Discover”, the mention to natural products was not removed since the approach A was used to the virtual screening of natural products from StreptomeDB. However the “Marine Natural Product” was changed to “Natural Product”.

II – Abstract

Line 24: NP hit drug discovery - remove 'hit'

 The abstract was changed accordingly.

 II – Introduction

“Lines 71-75: One more example is 'indolylidinepyrazolones' Tetrahedron Letters Volume 55, Issue 36, 3 September 2014, 5014-5018”

The indolylidinepyrazolone derivatives was added as well as the new reference 12:

 Indrasena, A.; Riyaz, S.; Malipeddi, P.L.; Padmaja, P.; Sridhar, B.; Dubey, P.K. Design, synthesis, and biological evaluation of indolylidinepyrazolones as potential anti-bacterial agents.              Tetrahedron Lett. 2014, 55, 5014–5018.

 III – Introduction

Line 110: check [10,29]-1)

 To avoid misinterpretation the text has been changed to “…and fungi were reported [10,30] ― (1) isolated from marine-derived bacteria: marinopyrrole A [31], an alkaloid endowed (Streptomyces sp.), and abyssomicin C [32], a spirotetronate poliketide (Verrucosispora sp.); (2) isolated from marine-derived fungus…

 IV – Results and Discussion

Lines 148-181: Can this be summarized into a table?

 Although we understand the concerns of reviewer 1, putting all these numbers in the Table 1 would be quite complicated, but we point out that although the minimum of pMIC in each cluster is not shown in the Table 1, the maximum is represented. On the other hand even if all these data were presented in the Table 1 it would always be necessary to analyze these data as it was done in the manuscript.

 IV – Results and Discussion

Why didn't you use the same datasets for both approaches and do a comparison?

 We thank the reviewer 1 for raising this issue, the data set of approach B cannot be used in approach A, because we do not known the chemical structure of all pure compounds and mixture of compounds in the fraction and extracts. However we can use the molecules of data set A in the approach B but we need to predict the 1H and 13C NMR spectra that would be huge effort. The development of QSAR model using predicted NMR spectra of molecules and mixture of molecules (simulating fractions and extracts) would be an important strategy to be developed in future work.

A new sentence was added to the Conclusion: “Moreover, the development of QSAR model using predicted NMR spectra of molecules and mixture of molecules (simulating fractions and extracts) would be an important strategy to be developed in future work.

Reviewer 2 Report

Disclaimer: I am a medicinal chemist and not a computational chemist. I am reviewing this article from the point of view of someone who would like to apply chemoinformatics approaches to discover enzyme inhibitors.

This is an interesting manuscript that describes the use of molecular and NMR descriptors to predict antibacterial activity MRSA. Machine learning and chemoinformatics are important emerging techniques and it is pleasing to see studies that explore their applications in drug discovery, especially in important targets such as MRSA.

Page 2 Lines 57-64: This paragraph does not make sense. Ribosomes, b-lactamases, etc are inhibition targets. They are not 'target-based approaches' or 'tools'. HTS (e.g. Nat Rev Drug Discov. 2011, 10, 188–195), fragment-based drug discovery (e.g. Molecules 2016, 21, 854), protein-directed dynamic combinatorial chemistry (e.g. Molecules 2016, 21, 910) are approaches!

It is interesting that NMR chemical shifts is a useful descriptor. I guess they reflect the structure of the molecule and the moiety of the molecule (e.g. hydrophobic aromtic groups vs polar NHs, etc). I find this manuscript a joy to read and, as a medicinal chemist who is keen to apply these approaches to discover new antibiotics, I think it will be of interest to the readers of Marine Drugs.

Author Response

Manuscript ID: biomolecules-321595

Title: "A Computer-Driven Approach to Discovery Marine Natural Product Leads for Methicillin-resistant Staphylococcus aureus Infection Therapy "

Author(s): Tiago Dias, Susana P. Gaudêncio and Florbela Pereira

Reviewer 2

Critical

I – Introduction

“Page 2 Lines 57-64: This paragraph does not make sense. Ribosomes, b-lactamases, etc are inhibition targets. They are not 'target-based approaches' or 'tools'. HTS (e.g. Nat Rev Drug Discov. 2011, 10, 188–195), fragment-based drug discovery (e.g. Molecules 2016, 21, 854), protein-directed dynamic combinatorial chemistry (e.g. Molecules 2016, 21, 910) are approaches!”

 Page 2, Lines 87-100, the sentences was changed to “Although target-based approaches using inhibition targets such as ribosome…